# Rethinking Aleatoric and Epistemic Uncertainty

**Freddie Bickford Smith**
University of Oxford

**Jannik Kossen**
University of Oxford

**Eleanor Trollope**
University of Oxford

**Mark van der Wilk**
University of Oxford

**Adam Foster**
University of Oxford

**Tom Rainforth**
University of Oxford

## Abstract

The ideas of aleatoric and epistemic uncertainty are widely used to reason about the probabilistic predictions of machine-learning models. We identify incoherence in existing discussions of these ideas and suggest this stems from the aleatoric-epistemic view being insufficiently expressive to capture all of the distinct quantities that researchers are interested in. To explain and address this we derive a simple delineation of different model-based uncertainties and the data-generating processes associated with training and evaluation. Using this in place of the aleatoric-epistemic view could produce clearer discourse as the field moves forward.

## 1 Introduction

When making decisions under uncertainty, it can be useful to reason about where that uncertainty comes from (Osband et al, 2023; Wen et al, 2022). Researchers commonly refer to the ideas of aleatoric (literal meaning: "relating to chance") and epistemic ("relating to knowledge") uncertainty, which have a long history in the study of probability (Hacking, 1975). Aleatoric uncertainty is typically associated with statistical dispersion in data or outcomes, while epistemic uncertainty is associated with the internal information state of a model (Hüllermeier & Waegeman, 2021).

Concerningly given their scale of use, these ideas are not being discussed coherently in the literature. The line between model-based predictions and data-generating processes is repeatedly blurred (Adlam et al, 2020; Amini et al, 2020; Ayhan & Berens, 2018; Collier et al, 2020; Immer et al, 2021; Kapoor et al, 2022; Liu et al, 2022; Mavor-Parker et al, 2022; Notin et al, 2021; Postels et al, 2019; Smith & Gal, 2018; van Amersfoort et al, 2020). Different mathematical quantities are used to refer to notionally the same concepts: epistemic uncertainty, for example, has received multiple definitions, including variance-based measures (Gal, 2016; Kendall & Gal, 2017; McAllister, 2016), information-based measures (Gal et al, 2017), ad-hoc reinterpretations of information-based measures (Shen et al, 2018; Siddhant & Lipton, 2018) and distance-based measures (Mukhoti et al, 2021, 2023; van Amersfoort et al, 2020). Misleading connections are drawn between predictive uncertainty and accuracy (Orlando et al, 2019; Wang et al, 2019). Tenuous assumptions are made about how predictive uncertainty will decompose on unseen data (Seeböck et al, 2019; Wang & Aitchison, 2021).

We suggest this incoherence arises from the aleatoric-epistemic view being too simplistic in the context of machine learning. To make this diagnosis precise and provide an alternative to the aleatoric-epistemic view, we systematically disambiguate some key concepts that arise in machine learning. We start with a predictive task of interest, highlight that the training data need not correspond directly to the task, and contrast model-based predictions with external data-generating processes.

This exposition supports our diagnosis of the issue in the discourse. We show that, like much of the literature that followed it, the popular interpretation of aleatoric and epistemic uncertainty presented in Gal (2016), Gal et al (2017) and Kendall & Gal (2017) is itself incoherent: it attaches multiple

Workshop on Bayesian Decision-making and Uncertainty, 38th Conference on Neural Information Processing Systems (NeurIPS 2024).

quantities to the same concepts. This conceptual overloading stems from attempting to encompass too many ideas within an uncertainty decomposition that has a fundamentally limited expressive capacity. Its effect is to conflate concepts that ought to be recognised as distinct.

Returning to foundational ideas from Bayesian statistics, we identify an alternative decomposition of predictive uncertainty that can be cleanly linked back to quantities used in past work. We believe this could be a basis for clearer thinking in future work, helping the field more quickly achieve its goals.

## 2   Background

The use of the terms "aleatoric" and "epistemic" in machine learning follows a history of use in the engineering literature. Special issues of *Reliability Engineering and System Safety* on "Treatment of aleatory and epistemic uncertainty" (Helton & Burmaster, 1996) and "Alternative representations of epistemic uncertainty" (Helton & Oberkampf, 2004) aggregated a considerable amount of discourse. More recent work includes that of Der Kiureghian & Ditlevsen (2009) and Helton et al (2010).

That literature itself builds on a much longer thread of work on sources of uncertainty. Helton & Oberkampf (2004) wrote that "this dual use of probability in the representation of both aleatory and epistemic uncertainty can be traced back to the beginning of the formal development of probability in the late 1600s (Bernstein, 1996; Hacking, 1975; Shafer, 1978)". Modern statistics texts referring to the ideas, even if not the exact terms, include the work of Chernoff & Moses (1959).

While the concepts of aleatoric and epistemic uncertainty had previously been used in machine learning, for example by Lawrence (2012) and Senge et al (2014), they were popularised by Gal (2016), Gal et al (2017) and Kendall & Gal (2017). The prevailing mathematical definitions are the information-theoretic quantities used by Gal et al (2017), building on earlier work on Bayesian experimental design (Lindley, 1956) and active learning (Houlsby et al, 2011; MacKay, 1992a,b).

## 3   Key concepts

Many quantities that arise in machine learning have been associated with the ideas of aleatoric and epistemic uncertainty. We set out to provide a clear synthesis of some key concepts.

**Reasoning should start with the predictive task of interest**   We consider prediction of $y|x$ where $x \in \mathcal{X}$ is an input and $y \in \mathcal{Y}$ is an output. We allow $x = \mathcal{X} = \emptyset$. This setup covers a wide range of scenarios, from predicting the bias of a coin ($\mathcal{X} = \emptyset$ and $\mathcal{Y} = [0, 1]$) to predicting the next word in a sentence ($\mathcal{X} = \mathcal{V}^l$ and $\mathcal{Y} = \mathcal{V}$ where $\mathcal{V}$ is a vocabulary and $l$ is the number of words so far).

**Training data need not correspond directly to the prediction**   Typically we have access to some training data, $d_{1:n} \sim p_{\text{train}}(d_{1:n})$, that can inform our prediction. It is common to assume $d_i \in \mathcal{X} \times \mathcal{Y}$. We emphasise that this is not necessary. The data could belong in some altogether separate space.

**Using a model allows data-driven prediction**   In machine learning we work from training data to predictions through a model, $p_n(y|x) = p(y|x; d_{1:n})$. Here we focus on parametric models. While not required for many of the quantities we consider, there can be stochastic parameters, $\theta \sim p_n(\theta) = p(\theta; d_{1:n})$, within the model, defined such that $p_n(y|x) = \mathbb{E}_{p_n(\theta)}[p(y|x, \theta)]$. We use $p_\infty(y|x)$ to denote the model we converge to as $n \to \infty$, which we assume is well defined.

**Predictions are generally distinct from data-generating processes**   Since we can have $d_i \notin \mathcal{X} \times \mathcal{Y}$, the link between $p_n(y|x)$ and $p_{\text{train}}(d_{1:n})$ can be weak. Suppose $d_{1:n}$ represents the outcomes of $n$ fair coin tosses and the task is to predict the coin's bias, so $\mathcal{X} = \emptyset$ and $\mathcal{Y} = [0, 1]$. Then as $n \to \infty$ the predictive entropy (Shannon, 1948), $\text{H}[p_n(y|x)]$, can tend to zero while $\text{H}[p_{\text{train}}(d_{1:n})] = n \log 2$ tends to infinity. Even if $d_i \in \mathcal{X} \times \mathcal{Y}$ and $d_i \sim p_{\text{train}}(y|x)p_{\text{train}}(x)$, it can still be the case that $p_n(y|x) \neq p_{\text{train}}(y|x)$ for all $n$ due to model misspecification (Kleijn & van der Vaart, 2006).

**Reference systems allow grounded evaluation**   Computing model-based uncertainties is not a general substitute for evaluating a model using a reference system, such as a person, physical sensor or computer program. Often this system performs the same predictive task as the model, and we can formalise evaluation as a comparison between the model and the reference system, $p_{\text{eval}}(y|x)$, on an input, $x$. If $x$ is considered to be drawn from some $p_{\text{eval}}(x)$, this commonly gives rise to expected losses of the form $\mathbb{E}_{p_{\text{eval}}(x,y)}[\ell(p_n(y|x), y)]$, often estimated using sampled $(x, y) \sim p_{\text{eval}}(x, y)$.

**Figure 1** A popular view on aleatoric and epistemic uncertainty attaches multiple quantities to each term. Some of these quantities can coincide in particular cases but in the general case they are distinct. The quotations here are from Kendall & Gal (2017); the interpretation of Equation 1 is due to Gal (2016) and Gal et al (2017).

## 4 Assessing a popular view

Having established some key concepts, we can now see how they connect to the aleatoric-epistemic view as presented in Gal (2016), Gal et al (2017) and Kendall & Gal (2017). The mathematical decomposition that has prevailed over time relates three information-theoretic quantities:

$$\underbrace{\mathrm{BALD}(x)}_{\text{epistemic}} = \underbrace{\mathrm{H}[p_n(y|x)]}_{\text{total}} - \underbrace{\mathbb{E}_{p_n(\theta)}[\mathrm{H}[p(y|x,\theta)]]}_{\text{aleatoric}}. \tag{1}$$

Gal (2016) stated the "total = aleatoric + epistemic" relationship and the correspondence between $p_n(y|x)$ and the total uncertainty, while Gal et al (2017) made the explicit link to Equation 1, informed by Houlsby et al (2011). Kendall & Gal (2017) expanded on these ideas in a computer-vision context.

Aleatoric and epistemic uncertainty as discussed in this work refer to multiple quantities (Figure 1), introducing a number of spurious associations. The competing definitions of aleatoric uncertainty conflate a model's irreducible predictive entropy, $\mathrm{H}[p_\infty(y|x)]$, with three separate quantities:

(a) $\mathrm{H}[p_{\mathrm{train}}(d_{1:n})]$, the entropy of the training data. Issue: because $d_i$ need not belong in $\mathcal{X} \times \mathcal{Y}$, the model's asymptotic predictive entropy can have little to do with $\mathrm{H}[p_{\mathrm{train}}(d_{1:n})]$.

(b) $\mathrm{H}[p_{\mathrm{eval}}(y|x)]$, the entropy of the reference system used in evaluation. Issue: we are not guaranteed to recover $p_{\mathrm{eval}}(y|x)$ as $n \to \infty$, for example if $p_{\mathrm{eval}}(y|x)$ is not in the model class.

(c) $\mathbb{E}_{p_n(\theta)}[\mathrm{H}[p(y|x,\theta)]]$, the conditional entropy of $y|x$ given the model parameters, $\theta \sim p_n(\theta)$. Issue: for finite $n$ the conditional entropy is only an estimator of $\mathrm{H}[p_\infty(y|x)]$ (Proposition 1).

Meanwhile the multiple definitions of epistemic uncertainty mix up a model's reducible predictive entropy, $\mathrm{H}[p_n(y|x)] - \mathrm{H}[p_\infty(y|x)]$, with two different quantities:

(a) $\mathrm{H}[p_n(\theta)]$, the entropy of the model parameters. Issue: predictions are often a non-invertible function of the parameters, and the parameter entropy at finite $n$ need not relate to $\mathrm{H}[p_\infty(y|x)]$.

(b) $\mathrm{H}[p_n(y|x)] - \mathbb{E}_{p_n(\theta)}[\mathrm{H}[p(y|x,\theta)]]$, the BALD score evaluated at $x$. Issue: for finite $n$ the BALD score is only an estimator of $\mathrm{H}[p_n(y|x)] - \mathrm{H}[p_\infty(y|x)]$ (Proposition 2).

Other sources of confusion in this view on aleatoric and epistemic uncertainty include an incorrect association between a model's subjective uncertainty and objective measures of performance, such as classification accuracy (Figure 2 in Kendall & Gal (2017)), along with misleading implications about how a model's uncertainty will behave with varying $n$ (Figure 6.11-6.12 in Gal (2016) and Table 3 in Kendall & Gal (2017)) and varying distance from the training data ("Aleatoric uncertainty does not increase for out-of-data examples... whereas epistemic uncertainty does" in Kendall & Gal (2017)).

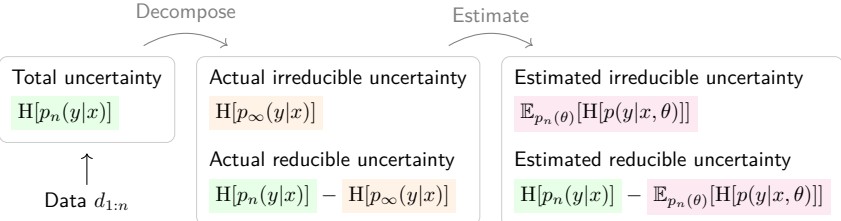

**Figure 2** The total predictive uncertainty (entropy) of a model, $p_n(y|x)$, trained on data $d_{1:n}$ can be decomposed into actual irreducible and reducible components, which reflect how the model's uncertainty changes as $n \to \infty$. Given finite $n$, these irreducible and reducible components must be estimated. This is made possible by using stochastic model parameters, $\theta \sim p_n(\theta)$, where $p_n(y|x) = \mathbb{E}_{p_n(\theta)}[p(y|x, \theta)]$.

## 5 An alternative perspective

Now we return to the goal of decomposing predictive uncertainty. If a model's prediction is uncertain, we want to know whether that prediction is fundamentally uncertain for the given model class or instead due to a lack of data. This breakdown has clear utility if our aim is to identify new data that will reduce a model's predictive uncertainty (Bickford Smith et al, 2023, 2024). But it is also relevant elsewhere. In model selection, for example, we might want to quantify a model's scope for improvement by forecasting how it will behave after training on more data.

We formalise this using a Bayesian perspective, which corresponds to reasoning about data that has not yet been observed (Fong et al, 2023). More concretely we revisit BALD's foundations in the framework of Bayesian experimental design (Lindley, 1956; Rainforth et al, 2024) and focus on the core idea of information gain. For a generic variable of interest, $\psi$, the information gain is defined as the reduction in entropy in $\psi$ that results from observing new data, $d_{(n+1):(n+m)}$:

$$\text{IG}_\psi(d_{(n+1):(n+m)}) = \text{H}[p_n(\psi)] - \text{H}[p_{n+m}(\psi)]. \tag{2}$$

Setting $\psi$ to $\theta$ and averaging over possible data recovers BALD (Houlsby et al, 2011) for $m = 1$ and BatchBALD (Kirsch et al, 2019) for $m > 1$. Here we are interested in the task of predicting $y|x$, so we instead set $\psi$ to $y|x$ (Bickford Smith et al, 2023). Considering the resulting information gain in the limit of infinite new data, $m \to \infty$, gives a decomposition of a model's total predictive entropy:

$$\underbrace{\text{IG}_{y|x}(d_{(n+1):\infty})}_{\text{reducible}} = \underbrace{\text{H}[p_n(y|x)]}_{\text{total}} - \underbrace{\text{H}[p_\infty(y|x)]}_{\text{irreducible}}. \tag{3}$$

In practice we do not have the new data used in this definition. Instead we have to reason about what the data could be, giving rise to estimators of the irreducible and reducible predictive entropy. Only in this context of estimation does a stochastic model become necessary: the decomposition in Equation 3 is based on a Bayesian perspective but is well defined for deterministic models.

**Proposition 1** *A model's conditional predictive entropy, $\mathbb{E}_{p_n(\theta)}[\text{H}[p(y|x, \theta)]]$, is a Bayes estimator of its irreducible predictive entropy, $\text{H}[p_\infty(y|x)]$.*

*Proof* Let $h$ be an estimator of $\text{H}[p_\infty(y|x)]$, and let $p_n(\theta)$ be our beliefs over which $\theta$ produces $p(y|x, \theta) = p_\infty(y|x)$. The Bayes estimator is the minimiser of the posterior expected loss, $L(h) = \mathbb{E}_{p_n(\theta)}[\ell(h, \theta)]$. If we use a quadratic loss, $\ell(h, \theta) = (h - \text{H}[p(y|x, \theta)])^2$, this minimiser satisfies

$$\nabla_h L(h) = \mathbb{E}_{p_n(\theta)}[2(h - \text{H}[p(y|x, \theta)])] = 0. \tag{4}$$

Solving this gives $h = \mathbb{E}_{p_n(\theta)}[\text{H}[p(y|x, \theta)]]$, the conditional predictive entropy. $\square$

**Proposition 2** *The BALD score evaluated at $x$, $\text{BALD}(x) = \text{EIG}_\theta(x)$, is a Bayes estimator of the reducible predictive entropy at $x$, $\text{IG}_{y|x}(d_{(n+1):\infty})$.*

*Proof* This follows directly from Equations 1 and 3 along with Proposition 1. $\square$

**Proposition 3** *The error associated with the approximation* $\mathrm{BALD}(x) \approx \mathrm{IG}_{y|x}(d_{(n+1):\infty})$ *corresponds to that associated with* $\mathbb{E}_{p_n(\theta)}[\mathrm{H}[p(y|x,\theta)]] \approx \mathrm{H}[p_\infty(y|x)]$.

*Proof* This again follows from Equations 1 and 3:

$$\mathrm{IG}_{y|x}(d_{(n+1):\infty}) - \mathrm{BALD}(x) = \mathbb{E}_{p_n(\theta)}[\mathrm{H}[p(y|x,\theta)]] - \mathrm{H}[p_\infty(y|x)]. \tag{5}$$

Both sides can be seen as approximation errors. □

There are cases where we can expect the estimators in Propositions 1 and 2 to be accurate. We might for example have $x, y \in \mathbb{R}$ and know that modelling $y|x$ as Gaussian-distributed with equal variance across all $x$ is appropriate, and we might even know exactly what variance to use. But in the general case these estimators can be highly inaccurate even if they are principled. Figure 2 in Bickford Smith et al (2024) demonstrates this: Bayesian deep learning can produce estimates of irreducible and reducible predictive entropy that are severely at odds with the entropy changes that occur in practice.

The conceptual map in Figure 2 combines the decomposition from Equation 3 with the estimators from Propositions 1 and 2. It thus connects Equation 3 back to Equation 1 while highlighting that the quantities in the latter should be seen not as direct measures of irreducible and reducible uncertainty but instead as estimators that might not be accurate. This new perspective gives us a clearer basis for reasoning about predictive uncertainty than is provided by the aleatoric-epistemic view.

## 6 Related work

A number of different perspectives on aleatoric and epistemic uncertainty in machine learning have been put forward in recent years. These include a discussion of where uncertainty comes from in machine learning (Gruber et al, 2023); a case against Shannon entropy for notions of predictive uncertainty (Wimmer et al, 2023); proposals for using alternative information-theoretic quantities (Schweighofer et al, 2023a,b, 2024); and various other suggestions for how to define uncertainty, such as in terms of frequentist risk (Lahlou et al, 2023), class-wise variance (Sale et al, 2023b, 2024b), credal sets (Hofman et al, 2024a; Sale et al, 2023a), distances between probability distributions (Sale et al, 2024a) and proper scoring rules (Hofman et al, 2024b). In contrast with most of that work, our approach here has been to consider the minimal changes needed to address the issues we have identified in existing discussions of aleatoric and epistemic uncertainty.

## 7 Conclusion

We have identified sources of confusion in the aleatoric-epistemic view on uncertainty in machine learning and, to deal with this, we have presented an alternative perspective. A key fact underlying this work is the extent of the subjectivity of the uncertainties we have discussed. Our presentation of the ideas makes transparent the dependence on the model class and the amount of data. But it abstracts away dependencies on other things, such as what exactly the data is and what algorithm is used to learn from the data. While we believe this abstraction is useful for the high-level understanding we have aimed to provide here, a precise analysis would need to account for these dependencies.

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
