# OpenReview forum: "Rethinking Aleatoric and Epistemic Uncertainty"
_NeurIPS.cc/2024/Workshop/BDU — NeurIPS BDU Workshop 2024 Oral_

### Official Review · Reviewer_onNc · 2024-09-24
**Good paper on estimation of aleatoric and epistemic uncertainty**

**Rating:** 9
**Confidence:** 5

**Review:**

This paper is a critique and alternate formulation for the common interpretation and estimation of aleatoric and epistemic uncertainty.

The authors argue that commonly used formulations for aleatoric and epistemic uncertainty are often wrong or contradictory, and they propose a partially new formulation (based on the well known information theoretical formulation for disentanglement). The paper only makes a theoretical argument. The paper is strong and should be accepted.

**Strengths**: The paper is very well written and argued, notation is very clear. The paper tackles an important open problem, as while there are very clear definitions for aleatoric and epistemic uncertainty (irreducible vs reducible uncertainties), there are many ways to estimate them, some more correct than others, and this paper make arguments on why some estimations are not completely correct, while providing a more correct formulation and a proof about estimating irreducible (aleatoric) uncertainty.

This paper is definitely interesting and would trigger interesting discussion in the workshop.

**Weaknesses**: The paper does not seem to have results, only the formulation, validation would have been nice, in particular with settings where training set size is varied.

---

### Official Review · Reviewer_ecq5 · 2024-09-30

**Rating:** 8
**Confidence:** 4

**Review:**

This paper notes systemic issues with standard definitions/decompositions of epistemic and aleatoric uncertainty as used in modern machine learning literature and proposes a new framework for properly decomposing uncertainty into reducible and irriducible elements. This framework is largely based on ideas from Fong's work on martingale posteriors as well as Rainforth's work on modern Bayesian experimental design.

This new framework is thought provoking and likely controversial. The authors' math and arguments are sound, correct, and rooted in modern literature. I imagine this submission will make for great discussion at the workshop and so I recommend that it is featured as a spotlight presentation.

---

### Decision · Program_Chairs · 2024-10-09

Accept (Oral)